# FAIRREWEIGHING: DENSITY ESTIMATION-BASED REWEIGHING FRAMEWORK FOR IMPROVING SEPARATION IN FAIR REGRESSION

## ABSTRACT

There has been a prevalence of implementing machine learning technologies in both high-stakes public-sector and industrial contexts. However, the lack of transparency in these algorithmic solutions has raised concerns over whether these data-informed decisions secure fairness against people from all racial, gender, or age groups. Despite the extensive research and work that emerged on fairness-aware machine learning, up till now, most efforts on solving this issue have been dedicated to binary classification tasks. In this work, we propose a generalized Reweighing pre-processing algorithm with density estimation to ensure the separation criterion $\hat{Y} \perp A \mid Y$ in fair regression problems. Evaluated by the ratio estimation of separation via probabilistic classification on both synthetic and real world data, we show that the proposed algorithm outperforms existing state-of-the-art regression fairness solutions in terms of maintaining high predicting accuracy while improving separation in fair regression.

## 1 INTRODUCTION

Over the last decade, we witnessed an explosion in the magnitude and variety of applications upon which statistical machine learning is exerted and practiced. The influence of such technology has drastically reshaped every aspect of our daily life, ranging from ads (Perlich et al., 2014) and movie recommendation engines (Biancalana et al., 2011) up to life-changing decisions that could lead to severe consequences. For instance, physicians and medical practitioners may determine if a patient should be discharged based on the likelihood that they possess the necessity of continuing hospitalization (Mortazavi et al., 2016). Similarly, police staffing and allocation might depend on future predictions of criminal activity across different communities (Redmond & Baveja, 2002). With its underlying power to affect risk assessment in critical judgment, we inevitably need to be cautious of the potential for such data-driven models to introduce or, in worse case, amplify existing discriminatory biases, thus becoming unfair. Journalists and academia have long discovered unintentional algorithmic unfairness lurking beneath the surface. For example, empirical findings demonstrate that AI can inherit the semantics biases humans exhibit in nature languages (Caliskan et al., 2017). Analysis of recidivism rate prediction also shows defendants of certain racial and ethnic groups are far more likely to be incorrectly assigned a higher risk of recidivism than other defendants (Chouldechova, 2017). Motivated by criticism and backlash from policymakers and primary media sources, the related research community has devoted tremendous effort to studying fairness and providing solutions that might improve our algorithms to avoid unfairness. Most of the studies have been focused on proposing and refining all sorts of mathematical definitions of fairness and algorithms for detecting and mitigating bias according to those metrics.

While fairness in binary classification settings has been extensively researched, where the predictions are discrete and categorical, researchers need to pay more attention to its counterpart in regression and ranking problems, where we are trying to estimate a continuous quantity. Most of the optimization on classifier-based systems is established within the context where the decisions are simply binary, for instance, college admission/rejection, credit card approval/decline, or whether a convicted individual would re-offend. However, in practice, many machine learning tools being utilized today merely serve as a reference to assist decision-makers in concluding. For example, hiring managers may consult a score-based model to rank applicants concerning their skills and capabilities, then decide

who should be interviewed based on their judgement (Chalfin et al., 2016). In these situations, classification models that provide a yes/no answer could deprive practitioners of their agency and autonomy (Sundar, 2020), whereas continuous scores only facilitate the decision-making process. Thus, it's essential to develop applicable optimization algorithms in regression settings.

Existing fairness metrics in fair regression literature are mostly adaptation of well-established mathematical formulations that was proposed for classification problems, such as Generalized Demographic Parity (GDP) (Jiang et al., 2022) and statistical parity (Agarwal et al., 2019). Although demographic parity guarantees that the predictions of a machine learning model are independent of the membership of a sensitive group, it also promotes laziness, where it solely concerns the model's outcome and does not ensure equal treatment. In addition, even if a classifier of regressor is perfectly accurate, it would still reflect some degree of bias as long as the actual class distribution varies among sensitive groups. Contrarily, separation — another measure of fairness that ensures model performs equally well for different groups — allows a perfect predictor when base rates are different Hardt et al. (2016). It also penalizes the laziness of demographic parity as it incentivizes the reduction of errors equally across all groups.

Therefore, in this paper, we draw inspiration from separation in classification setting and approximate density ratio estimation that are proposed for ranking and regression models (Steinberg et al., 2020). We develop a density estimation-based training framework that achieves fairness by adjusting the influence of each data entry, named *FairReweighing*. In addition, we demonstrate that the exact mechanism can support both classification and regression problems. By competing against state-of-the-art algorithms on fair regression, we highlight the effectiveness of our approach through comprehensive experiments on both synthetic and real-world data sets, intending to answer the following research questions:

- **RQ1: What is the performance of *FairReweighing* compared to state-of-the-art bias mitigation techniques in regression problems?**
- **RQ2: Is *FairReweighing* performing the same as *Reweighing* in classification problems?**

Our major contributions are as follows:

- We proposed a pre-processing technique *FairReweighing* specifically designed to satisfy the separation criterion for regression problems.
- Our approach demonstrates better performance against state-of-the-art mechanisms on both synthetic and real-world regression data sets.
- We also showed that the proposed *FairReweighing* algorithm can be reduced to the well-known algorithm *Reweighing* in classification problems. Therefore *FairReweighing* is a generalized version of *Reweighing* and it can be applied to both classification and regression problems.

## 2 RELATED WORK

### 2.1 FAIR CLASSIFICATION

Most of the existing studies in binary classification settings fall into two categories: suggesting novel fairness definitions to evaluate machine learning models and designing advanced procedures to minimize discrimination without much sacrifice on performance. Many fairness definitions for classification can be statistically calculated to compare and improve. *Unawareness* (Grgic-Hlaca et al., 2016) demands any decision-making process to exclude the sensitive attribute in the training process. Yet, it suffers from the inference of sensitive characteristics through unprotected traits that serve as proxies (Corbett-Davies & Goel, 2018). Group fairness is a collection of statistical measurements centering on whether some pre-defined metrics of accuracy are equal across different groups divided by the protected attributes. *Demographic parity* (Dwork et al., 2012) ensures that the positive rate for the target variable is the same across people belonging to different demographic groups. *Separation* is defined as the condition where the sensitive attributes $A$ are conditionally independent of the prediction $\hat{Y}$, given the target value $Y$, and we write:

$$\hat{Y} \perp A \mid Y \qquad \text{Equivalently,} \qquad P(\hat{Y} \mid A, Y) = P(\hat{Y} \mid Y) \qquad (1)$$

For a binary classifier, achieving separation is equivalent to imposing the following two constraints:

$$P(\hat{Y} = 1 \mid Y = 1, A = a) = P(\hat{Y} = 1 \mid Y = 1, A = b) \tag{2}$$

$$P(\hat{Y} = 1 \mid Y = 0, A = a) = P(\hat{Y} = 1 \mid Y = 0, A = b) \tag{3}$$

Recall that $P(\hat{Y} = 1 \mid Y = 1)$ is referred to as the true positive rate, and $P(\hat{Y} = 1 \mid Y = 0)$ as the false positive rate of the classifier. Essentially, *separation* or *equalized odds* (Hardt et al., 2016) requires that subjects in protected and unprotected groups have equal true positive rates and false positive rates. On the other hand, individual fairness considers the treatments between pairs of individuals rather than comparing at the group level. It was proposed under the principle that "similar individuals should be treated similarly" (Dwork et al., 2012). Individual fairness is more fine-grained and meaningful than statistical definitions in terms of semantics and interpretability. However, because the notion is established on assumptions of the relationship between features and labels, it is hard to find appropriate metrics to compare two individuals (Chouldechova & Roth, 2020).

Based on these metrics, researchers have developed a variety of pre-processing algorithms and frameworks dedicated to modifying the training set and overcoming data imbalance while attempting to alter as little as possible. The underlying assumption is that the unfairness we observed in machine learning models may already include discrimination rooted in training data. Jiang & Nachum (2020) shows that you can train an unbiased machine learning classifier on the re-weighted dataset with unobserved but unbiased labels. Feldman et al. (2015) propose a method to test and eliminate disparate impact while preserving important information in the data. Zemel et al. (2013) achieve both group and individual fairness by introducing an intermediate representation of the data and obscuring information on sensitive attributes. *FairBalance* proposed by Yu et al. (2021) balance training data distribution across every protect feature to support group fairness. And above all, Kamiran & Calders (2012) propose *reweighing* by carefully adjusting the influence of the tuples in training set to be discrimination-free without altering ground-truth labels. In this paper, we extend this mechanism to work on classification and regression models with proper weight selection and assigning.

## 2.2 Fair Regression

Defining fairness metrics in regression settings has always been challenging. Unlike classification, no consensus on its formulation has yet emerged. The significant difference from the prior classification setup is that the target variable is now allowed to be continuous rather than binary or categorical.

### 2.2.1 Fairness notions for regression problems

Berk et al. (2017) introduce a flexible family of pairwise fairness regularizers for regression problems based on individual notions of fairness in classification settings where similar individuals should be treated similarly. Agarwal et al. (2019) propose general frameworks for achieving fairness in regression under two fairness criteria, statistical parity, which requires that the prediction is statistically independent of the protected attribute, and bounded group loss, which demands that the prediction error for any protected group does not exceed a pre-defined threshold.

Jiang et al. (2022) present Generalized Demographic Parity (GDP) as a fairness metric for both continuous and discrete attributes that preserve tractable computation, and justify its unification with the well-established demographic parity. However, it also inherits its disadvantage of ruling out a perfect predictor and promoting laziness when there is a causal relationship between the protected attribute and the output variable. Another recent work by Narasimhan et al. (2020) proposes a collection of group-dependent pairwise fairness metrics for ranking and regression models that resembles the notion of *Average odds difference* for fair classification. They translate classification-exclusive statistics like true positive rate and false positive rate into the likelihood of correctly ranking pairs and develop corresponding metrics.

Based off separation in classification setting, Steinberg et al. (2020) present methods for efficiently and numerically approximating the separation group fairness criteria within the broader context of regression. To begin, they initially reformulate the fairness criteria in the form of density ratios,

$$\mathbb{E}_{\hat{Y}}[r_{sep}] = \frac{P(\hat{Y} \mid A = 1, Y)}{P(\hat{Y} \mid A = 0, Y)}, \tag{4}$$

where a ratio of 1 would represent the most fair. Empirically calculating the ratio is a common approach for assessing the fairness criteria in classification. However, in a regression context, it is notably simpler to directly estimate the ratios using *density ratio estimation* methods, rather than separately estimating the probability densities in equation and subsequently computing the ratio (Sugiyama et al., 2010). A widely recognized approach involves directly approximating density ratios using the output of a probabilistic classifier. In such scenarios, we can employ the following:

$$P(A = a \mid \hat{Y} = \hat{y}) \approx \rho(a \mid \hat{y}) \tag{5}$$

$$P(A = a \mid Y = y, \hat{Y} = \hat{y}) \approx \rho(a \mid y, \hat{y}) \tag{6}$$

Here, $\rho(a \mid \hat{y})$ and $\rho(a \mid y, \hat{y})$ represents the prediction of the probability where $A = a$, which is generated by introducing and training two machine learning classifiers. Now, we can use such density ratio estimates to approximate $r_{sep}$ by using Bayes' rule, 5 and 6

$$\hat{r}_{sep} = \frac{1}{n} \sum_{i=1}^{n} \frac{\rho(1 \mid y_i, \hat{y}_i)}{1 - \rho(1 \mid y_i, \hat{y}_i)} \cdot \frac{1 - \rho(1 \mid y_i)}{\rho(1 \mid y_i)} \tag{7}$$

The approximation is easily interpreted: the separation approximations gauge the additional predictive power that the joint distribution of $Y$ and $\hat{Y}$ provides in determining $A$, compared to solely considering the marginal distributions of $Y$. These techniques are designed to be versatile, working independently of the specific regression algorithms employed.

In this paper, we focus on the ratio of separation metric in equation 7 since individual fairness Dwork et al. (2012) requires similar individuals (from different sensitive groups) to be treated similarly, demographic parity Dwork et al. (2012) requires the acceptance rates to be the same across different sensitive groups. On the other hand, the separation criterion always allows the perfect predictor to be evaluated as fair Hardt et al. (2016).

### 2.2.2 TREATMENTS FOR DIFFERENT REGRESSION FAIRNESS NOTIONS

Categorized as in-processing approaches, Berk et al. Berk et al. (2017), Agarwal et al. Agarwal et al. (2019) and Jiang et al. Jiang et al. (2022) all introduce a regularization (or penalty) term in the training process and aim to find the model which minimizes the associated regularized loss function, where Narasimhan et al. Narasimhan et al. (2020) directly enforce a desired pairwise fairness criterion using constrained optimization. Calders et al. (2013) also proposed a propensity score based stratification approach for balancing the dataset to address discrimination-aware regression. They define two measures for quantifying attribute effects and then develop constrained linear regression models for controlling the effect of such attribute on the predictions. It's essentially an in-processing mechanism that solve an optimization problem between predicting accuracy and discrimination.

In this paper, we propose (in the next section) a pre-processing technique to satisfy the separation criterion. By just reweighing the training data before training any regression model on it, we show that separation can be satisfied. This technique is more efficient and has significantly low overhead, competing against other mechanisms, which we will demonstrate later in our experiments. It is also considered more generalized as there is no limitation on which learning algorithms should be applied. This approach is free of trade-off or *Price of Fairness* parameters (weights) and can achieve competitively better balance between protected attributes and prediction accuracy.

## 3 FAIRREWEIGHING

**Problem Statement.** Consider training a model on a dataset $(X \in \mathbb{R}, A \in \{0, 1\}, Y \in \mathbb{R})$, the model will make predictions of $\hat{Y}$ based on its inputs $(x, a)$. The goal is to have the model's prediction $\hat{Y}$ satisfy the separation criterion in equation 1.

### 3.1 How FairReweighing ensures separation

To check whether separation is satisfied for the model on its training data, we have

$$P(\hat{Y} = \hat{y}|y) = \sum_a \sum_x P(\hat{Y} = \hat{y}|x, a)P(x, a|y)$$
$$P(\hat{Y} = \hat{y}|a, y) = \sum_x P(\hat{Y} = \hat{y}|x, a)P(x|a, y). \tag{8}$$

Under conditional independence assumption of the data $X \perp A|Y$, we have

$$P(\hat{Y} = \hat{y}|y) = \sum_a \sum_x P(\hat{Y} = \hat{y}|x, a)P(x|y)P(a|y)$$
$$P(\hat{Y} = \hat{y}|a, y) = \sum_x P(\hat{Y} = \hat{y}|x, a)P(x|y). \tag{9}$$

When the sensitive attribute has no impact on the dependent variable given other independent variables— $A \perp Y|X$, we can show that the model directly trained on the original training data satisfies the separation criterion:

$$P(\hat{Y} = \hat{y}|x, a) = P(Y = \hat{y}|x, a) = P(\hat{y}|x). \tag{10}$$

Apply equation 10 to equation 9 we have

$$P(\hat{Y} = \hat{y}|a, y) = \sum_x P(\hat{y}|x)P(x|y)$$
$$P(\hat{Y} = \hat{y}|y) = \sum_a \sum_x P(\hat{y}|x)P(x|y)P(a|y) = \sum_x P(\hat{y}|x)P(x|y) \sum_a P(a|y) \tag{11}$$
$$= \sum_x P(\hat{y}|x)P(x|y) = P(\hat{Y} = \hat{y}|a, y).$$

Therefore we have $\hat{Y} \perp A|Y$ and the model satisfies separation. However, often, the sensitive attribute does have an impact on the dependent variable given other independent variables— $A \not\perp Y|X$. Here we propose that, preprocessing the training data with FairReweighing will ensure separation $\hat{Y} \perp A|Y$ when $A \not\perp Y|X$. Extended upon the binary classification version of Reweighing algorithm in Kamiran & Calders (2012), FairReweighing adds a sample weight to each training data point:

$$W(a, y) = \frac{P(a)P(y)}{P(a, y)}. \tag{12}$$

After FairReweighing, the model learns from the weighted training data:

$$P(\hat{Y} = \hat{y}|x, a) = \frac{P(\hat{y}, x, a)}{P(x, a)} = \frac{P(\hat{y}, x, a)W(a, \hat{y})}{\sum_y P(y, x, a)W(a, y)}. \tag{13}$$

With equation 12 and the independence assumption of $X \perp A|Y$, we have

$$P(y, x, a)W(a, y) = \frac{P(a)P(y)P(y, x, a)}{P(a, y)} = P(a)P(y)P(x|a, y) = P(a)P(y)P(x|y) \tag{14}$$
$$= P(a)P(x, y).$$

Apply equation 14 to equation 13 we have

$$P(\hat{Y} = \hat{y}|x, a) = \frac{P(a)P(x, \hat{y})}{\sum_y P(a)P(x, y)} = \frac{P(x, \hat{y})}{\sum_y P(x, y)} = \frac{P(x, y)}{P(x)} = P(\hat{y}|x). \tag{15}$$

Therefore, FairReweighing ensures the conditional independence of $\hat{Y} \perp A|X$ and thus will lead to $\hat{Y} \perp A|Y$ as shown in equation 11. To sum up, preprocessing the training data with FairReweighing as equation 12 will ensure separation of the learned model under the condition that $X \perp A|Y$.

## 3.2 CALCULATION OF THE PREPROCESSING WEIGHTS IN FAIRREWEIGHING

To calculate the weight of FairReweighing in equation 12, this probability of $P(a)$, $P(y)$, and $P(a, y)$ can be easily translated into frequency counts in the binary classification setting. However, it will be much harder to obtain the underlying probability density for regression problems where both the target variable and protected feature can be continuous, and the same value may only occur once in the training data. One solution is to specify an interval threshold and segment the training data into a set of $N$ brackets. Then we can approximate it into a multiclass classification problem with $y_1...y_N$ target groups. However, such binning or bucketing mechanism throws down another challenge on where the lines between bins should be drawn, and it might introduce hyperparameters that need to be tuned manually. To accurately capture the density of the target variable and protected feature, we adopt two different nonparametric probability density estimation techniques.

**FairReweighing (Neighbor):** The first approach, radius neighbors (Goldberger et al., 2004), is an enhancement to the well-known k-nearest neighbors algorithm that evaluates density by taking into account all the samples within a defined radius of a new instance instead of just its k closest neighbors. By specifying the radius range and the metric to use for distance computation, we locate all neighbors of a given sample and use that as an estimation of its density:

$$\rho_N(x) = N(x_i \mid dist(x, x_i) < r) \quad \forall i \in \{1...N\} \tag{16}$$

**FairReweighing (Kernel):** Another method we implement is kernel density estimation (Terrell & Scott, 1992), which is a widely used nonparametric method for estimating the probability density function of a continuous random variable.

No matter which density estimation is administered through the pipeline, *FairReweighing* is a generalized algorithm that can be applied to both classification and regression problems. Furthermore, we might want to constrain multiple protected features in specific applications. For example, we may wish to optimize for fairness concerning gender and race simultaneously. Because both radius neighbors and kernel density estimation can be performed in any number of dimensions, our approach would automatically calculate the extent to which each feature must be reweighted to meet the fairness requirements. After the imbalance between the observed and expected probability density has been equalized as in equation 12, we use the calculated weights on our training data to learn for a discrimination-free regressor or classifier. The procedure outlining our *FairReweighing* method is detailed in Algorithm 1.

---

**Algorithm 1:** *FairReweighing*

**Input** : $(X \in \mathbb{R}, A \in \{0, 1\}, Y \in \mathbb{R})$, Given
**Output** : Regressor or classifier learned on reweighted $(X, A, Y)$

1 **for** $a_i \in A$ **do**
2     **for** $y_i \in Y$ **do**
3        $W(a_i, y_i) = \frac{\rho(a_i) \times \rho(y_i)}{\rho(a_i, y_i)}$
4     **end**
5 **end**
6 **for** $x \in X$ **do**
7     $W(x) = W(x(A), x(Y))$
8 **end**
9 Train a regressor or classifier on $(X, A, Y)$ allowing for weights $W(X)$
10 **return** Regressor or Classifier $M$

---

## 4 EXPERIMENT

In this section, we answer our research questions through experiments on both synthetic and real world data in both classification and regression settings. For radius neighbors, we find the best radius at 0.5 through cross-validation and calculate the euclidean distance between the target sample and other data points. When implementing KDE, we first dtandardize features by removing the mean and

scaling to unit variance, then proceed with gaussian as kernel function and bandwidth of 0.2 from a rule-of-thumb bandwidth estimator. Logistic regression models are being fitted in classification settings while linear regression models in regression[1]. Each data set is divided randomly into training and test sets with a ratio of 50:50, and we report the average of any given metric over 20 independent iterations. We compare **FairReweighing** against (a) Berk et al. (2017) which is designed to achieve convex individual fairness; (b) (Agarwal et al., 2019) which is designed to achieve statistical (or demographic) parity (DP) in fair classification and regression; and (c) Narasimhan et al. (2020) which is designed to achieve a pairwise fairness metric. Apart from comparing against these state-of-the-art bias mitigation techniques, we also include a none-treatment method as a baseline in all experiments.

## 4.1 REGRESSION

In order to answer **RQ1**, we study how *FairReweighing* (with a linear regression model trained on the pre-processed data) performs against the state-of-the-art bias mitigation algorithms for regression. Every treatment is evaluated by the $\hat{r}_{sep}$ metric from equation 7— the closer $\hat{r}_{sep}$ is to 1.0, the better the model satisfies separation. We conducted comprehensive experiments on one synthetic data set and two real world data sets.

### 4.1.1 SYNTHETIC DATA

Our synthetic data set is constructed as the right. The following are the critical assumptions for generating this data: vertical jump height is primarily determined by the height of the tester and his/her power level, and women tend to be shorter and have less power level than men on average. Our task is to predict a subject's vertical jump height

$$\text{gender}_i \sim Bernoulli(0.7)$$
$$\text{height}_i \mid \text{gender}_i = 1 \sim Normal(1.75, 0.15)$$
$$\text{height}_i \mid \text{gender}_i = 0 \sim Normal(1.65, 0.1)$$
$$\text{power}_i \mid \text{gender}_i = 1 \sim Normal(0.6, 0.15)$$
$$\text{power}_i \mid \text{gender}_i = 0 \sim Normal(0.5, 0.1)$$
$$\text{jump} \sim \text{height} + \text{power}$$

based on only gender and height (power as a hidden feature) and evaluate proposed fairness metrics on protected feature. The reason we construct such a naive data set is to test the accuracy of our density estimation methods by comparing the estimations with the ground truth distributions.

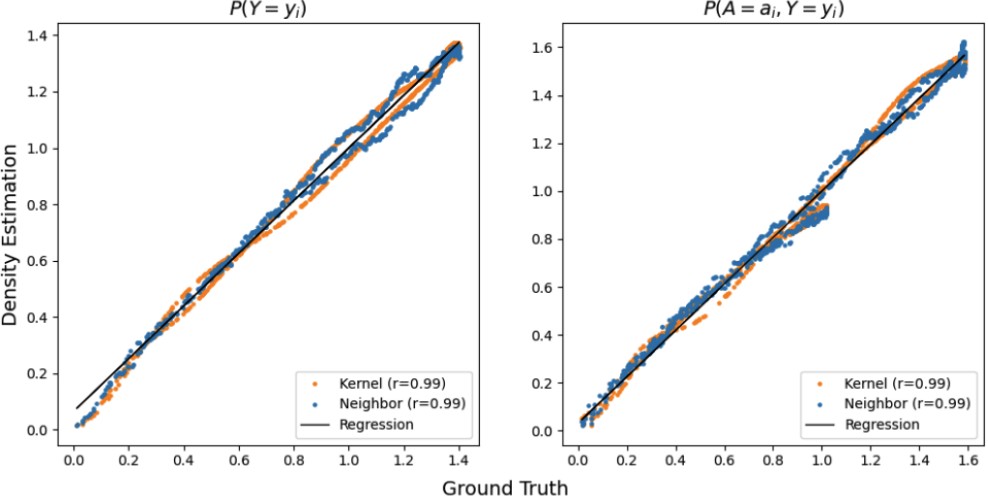

Figure 1: Comparison between density estimations and the ground truth distribution.

Figure 1 demonstrates the relationship between the two density estimations for $P(Y = y_i)$ and $P(A = a_i, Y = y_i)$ with their corresponding ground truth distributions. Both estimations have a strong positive correlation with Pearson correlation coefficient close to 1 ($r = 0.99$). Table 1 shows the accuracy and fairness metric when implementing *FairReweighing* with either ground truth distribution or density estimation approximation. The result suggests that *FairReweighing* with both density estimation successively improve separation while maintaining good prediction accuracy.

---

[1]Code is available at: https://anonymous.4open.science/r/FairReweighing-EC68

Table 1: Results on the synthetic regression problem.

| Data | Group | Models | MSE | R-Squared | $\hat{r}_{sep}$ |
|------|-------|--------|-----|-----------|-----------------|
| *Synthetic* | Gender | None | 0.018 | 0.606 | 2.014 |
| | | *FairReweighing* (Ground Truth) | 0.020 | 0.566 | 1.027 |
| | | *FairReweighing* (Neighbor) | 0.020 | 0.544 | 1.083 |
| | | *FairReweighing* (Kernel) | 0.019 | 0.585 | 1.090 |

### 4.1.2 REAL WORLD DATA

- The *Communities and Crimes* (Redmond & Baveja, 2002) data contains 1,994 instances of socio-economic, law enforcement, and crime data about communities in the United States. There are nearly 140 features, and the task is to predict each community's per capita crime rate. Practically, when the prediction of crime rate from the regression model trained on this data is being used to decide the necessary police strength in each community, ethical issue may arise when separation criterion is not satisfied for the model's prediction and whether the community is white dominant or not. This means, the prediction are incorrectly affected by the sensitive attribute for communities with the same crime rates. We will show in our experiment how separation will be evaluated and tackled by FairReweighing.

- The *Law School* (Wightman, 1998) data records information of 22,407 law school students, and we are trying to predict a student's normalized first year GPA based on his/her LSAT score, family income, full-time status, race, gender and the law school cluster the student belongs to. In our experiment, we treat race as the discrete protected attribute, divided into black and white students. Practically, when the prediction of first year GPA from the regression model trained on this data is being used to decide whether a student should be admitted, ethical issue may arise when separation criterion is not satisfied for the model's prediction and gender or race. This means, the prediction are incorrectly affected by these sensitive attributes for students with same first year GPAs. We will show in our experiment how separation will be evaluated and tackled by FairReweighing.

Table 2: Results on real world regression problems.

| Data | Group | Models | MSE | R-Squared | $\hat{r}_{sep}$ |
|------|-------|--------|-----|-----------|-----------------|
| *Crimes* | Race | None | 0.020 | 0.626 | 1.563 |
| | | Berk et al. | 0.024 | 0.553 | 1.130 |
| | | Agarwal et al. | 0.024 | 0.548 | 1.134 |
| | | Narasimhan et al. | 0.029 | 0.474 | 1.166 |
| | | *FairReweighing* (Neighbor) | 0.024 | 0.559 | 1.095 |
| | | *FairReweighing* (Kernel) | 0.024 | 0.562 | 1.091 |
| *Law School* | Race | None | 0.073 | 0.540 | 1.153 |
| | | Berk et al. | 0.075 | 0.518 | 1.096 |
| | | Agarwal et al. | 0.075 | 0.522 | 1.099 |
| | | Narasimhan et al. | 0.073 | 0.533 | 1.113 |
| | | *FairReweighing* (Neighbor) | 0.074 | 0.539 | 1.038 |
| | | *FairReweighing* (Kernel) | 0.074 | 0.551 | 1.034 |

### 4.1.3 RESULTS (**RQ1**)

Table 1 and Table 2 show the average accuracy and fairness metric on data sets in the regression setting with both synthetic and real world dataset. Predictions generated by the model trained after *FairReweighing* are significantly less biased in terms of $\hat{r}_{sep}$ against other treatments for both density estimation method. Across all three data sets, we see an average of 71% reduction in $\hat{r}_{sep}$, compared to all other treatment. In terms of accuracy, our proposed treatments retain high prediction accuracy in all cases, with MSE and R-Squared scores comparable to the non-treatment baseline. Although we did not fully minimize $\hat{r}_{sep}$ to 1 (no discrimination at all) in some cases, the results still demonstrate that our approach performs better than the state-of-the-art bias mitigation technique in terms of $\hat{r}_{sep}$ and can always consistently balances accuracy and fairness. It is also worth mentioning that none of the baselines compared in Table 2 were specifically designed to optimize for $\hat{r}_{sep}$. While Berk

et al. (2017) has been used to improve $\hat{r}_{sep}$ in Steinberg et al. (2020), it was originally designed to optimize for individual fairness. This also explains why FairReweighing achieves better separation when compared to other baselines— it is the first treatment specifically designed to optimize for separation in regression problems.

## 4.2 CLASSIFICATION

### 4.2.1 DATA SETS

In order to answer **RQ2**, we used two publicly available data sets in fair classification literature that assume binary protected groups:

- The *German Credit* (Dua & Graff, 2017) data consists of 1000 instances, with 30% of them being positively classified into good credit risks. There are nine features, and the task is to predict an individual's creditworthiness based on his/her economic circumstances. Gender is considered the protected attribute in our experiments, divided into two groups: female (31%) and male.
- The *Heart Disease* (Dua & Graff, 2017) data consists of 1190 instances with 14 attributes. The major task of this data set is to predict whether there is the presence of heart disease or not (0 for no disease, 1 for disease) based on the given attributes of a patient. In our experiments, gender is treated as the protected attribute.

Table 3: Results on real world classification problems.

| Data | Group | Models | Accuracy | F1 | $\hat{r}_{sep}$ | AOD |
|------|-------|--------|----------|-----|-----------------|-----|
| *German Credit* | Gender | None | 0.622 | 0.682 | 1.059 | 0.153 |
| | | *Reweighing* | 0.624 | 0.688 | 1.016 | 0.021 |
| | | *FairReweighing*(Neighbor) | 0.624 | 0.688 | 1.016 | 0.021 |
| | | *FairReweighing*(Kernel) | 0.624 | 0.688 | 1.016 | 0.021 |
| *Heart Disease* | Gender | None | 0.812 | 0.834 | 1.059 | -0.110 |
| | | *Reweighing* | 0.815 | 0.838 | 1.016 | 0.010 |
| | | *FairReweighing*(Neighbor) | 0.815 | 0.838 | 1.016 | 0.010 |
| | | *FairReweighing*(Kernel) | 0.815 | 0.838 | 1.016 | 0.010 |

### 4.2.2 RESULTS (**RQ2**)

The average results of 20 independent trials for classification are presented in Table 3.

- Here in this table, we used two metrics $\hat{r}_{sep}$ and Average Odds Difference (AOD) to measure separation in classification problems. AOD is a metric measuring the violation of equalized odds and the closer it is to 0, the better the model satisfies separation. Table 3 shows that $\hat{r}_{sep}$ and AOD are always consistent. This validates that, $\hat{r}_{sep}$ is applicable to measure separation for classification.
- We can also observe that both $\hat{r}_{sep}$ and AOD show the same trend— after applying either *Reweighing* or *FairReweighing*, separation will be better satisfied while maintaining prediction accuracy. More specifically, every metric is the same for *Reweighing* and *FairReweighing* in classification problems. This validates that, *FairReweighing* is identical with *Reweighing* for classification.

## 5 CONCLUSION

Ever since the discovery of underlying discriminatory issues with data-driven methods, the great majority of work in research communities on fairness in machine learning has centered around classic classification problems, where the target variable is either binary or categorical. However, it is only one facet of how machine learning is utilized. In this work, we introduced a pre-processing algorithm *FairReweighing* specifically designed to achieve the separation criterion in regression problems. Through comprehensive experiments on synthetic and real-world data sets, *FairReweighing* outperforms existing state-of-the-art algorithms in terms of satisfying separation for regression. We also show that, *FairReweighing* is reduced to the well-known pre-processing algorithm *Reweighing* when applied to classification problems.

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
