# OpenReview forum: "FairReweighing: density estimation-based reweighing framework for improving separation in fair regression"
_ICLR.cc/2024/Conference — Submitted to ICLR 2024_

### Official Review · Reviewer_gSA5 · 2023-10-17

**Soundness:** 3 good
**Presentation:** 1 poor
**Contribution:** 1 poor
**Rating:** 3
**Confidence:** 4

**Summary:**

This paper proposes a bias mitigation method called FairReweighing for regression tasks. FairReweighing extends the preprocessing approach by Kamiran & Calders for classification to also work for regression by using k-nearest neigbors or kernel density estimation instead of frequenty counting to estimate density. Experiments show that model trained after using FairReweighing are significantly less biased.

**Strengths:**

* Unfairness mitigation for regression is indeed relatively understudied.
* The fair reweighing extension seems simple and practical.

**Weaknesses:**

* The technical contribution seems marginal. The core issue seems to be how to estimate density. Compared to a classification setup where we can use frequency counts, the idea is to use k-nearest neighbors or kernel density estimation instead for regression. This looks like a straightforward generalization of Kamiran & Calders, but not substantial enough for a full paper.

* It is not clear why RQ2 is important. FairReweighing's goal is to improve fairness for regression, so why is it critical to validate its performance on classification problems? Instead, there should be other important research questions regarding regression. For example, how does pre-processing approaches compare to or complement in-processing approaches? How efficient is the unfairness mitigation? And so on.

* It is not clear why the inconsistency in (8) leads to unfairness as described in the paper. The authors say that the model would "prioritize these data points and produce more accurate predictions while disregarding" the other examples, but there does not seem to be any theoretical or empirical evidence. Therefore, it is difficult to understand why the weighting in (9) solves the problem.

* Satisfying a single fairness measure seems a bit limited, and the contributions would be stronger if other fairness measures in the literature proposed by Jiang et al. (2022) and Narasimhan et al. (2020) are also supported.

* It is not clear how accurate the approximation in (7) is.

* In Section 4.1.1, concluding that the density estimation approximation is accurate only based on synthetic data is not convincing.

* In Sections 4.1.2 and 4.2, only using two real datasets for the performance comparisons does not seem extensive enough.

**Questions:**

Same as the weak points above.

---

> ### Author Response · Authors · 2023-11-22
>
> First, we would like to thank the reviewer for the detailed comments. We have carefully considered all the comments and we believe that some of the comments have driven us to improve the paper greatly.
>
> Weakness 1. We have updated the contributions at the end of Introduction to highlight that, one major contribution is that, we now show in Section 3.1, FairReweighing can ensure separation in regression problems under the condition that $X\perp A | Y$. Before this paper, there is no treatment specifically designed for separation in regression problems.
>
> Weakness 2. RQ2 is explored to validate
> 1) FairReweighing is a generalized version of Reweighing;
> 2) Applying FairReweighing on classification is the same as applying Reweighing. RQ2 is indeed less important than RQ1 for this paper’s topic. But it serves as a validation of the generalizability of FairReweighing.
>
> Weakness 3. We would like to thank the reviewer for pointing out that there were not enough details on how FairReweighing is related to separation. This is why we have added a new Section 3.1 to show how FairReweighing ensures separation of the learned model under a conditional independence assumption of $X\perp A | Y$. We also demonstrate the effectiveness of our approach through empirical evidence on real world datasets.
>
> Weakness 4. As we have shown in Section 3.1, FairReweighing is specifically designed to satisfy separation in regression problems. As a result, there is no reason to apply it for other fairness notions. We also kindly point out that fairness notions are often contradictory to each other and are impossible to be satisfied at the same time. (*Friedler, Sorelle A., Carlos Scheidegger, and Suresh Venkatasubramanian. "The (im) possibility of fairness: Different value systems require different mechanisms for fair decision making." Communications of the ACM 64, no. 4 (2021): 136-143.*) Yes, there are in-processing algorithms which can be used to optimize for different fairness notions, however as shown in Table 2, none of those can satisfy separation as good as FairReweighing.
>
> Weakness 5 & 6. Since both KDE and radius neighbors are well-established density estimation algorithms, we did not provide the theoretical analyses of the accuracy of these algorithms. We would kindly ask the readers to refer to their original literature for the theoretical analyses. On the other hand, we provide some empirical results on a synthetic dataset in Figure 1 to show how accurate the estimations are when compared to the known ground truth distributions. Such comparison is impossible to make on real world datasets since we do not know the actual distributions of the variables.
>
> Weakness 7. We agree that experimenting on two real world datasets to provide validation for practical application scenarios does not seem extensive enough. However, we believe that both datasets we selected are sufficiently complex and diverse in nature and the method we proposed in the paper is indeed demonstrated to be effective in improving fairness. Meanwhile, we also kindly point out that, unlike the abundant number of classification fairness datasets, there are not many regression fairness datasets available to experiment on. For future work, we’ll continue our evaluation on datasets of all variety and complexity to provide more context on practical application scenarios.
>
> Again, we want to thank the reviewer for the detailed comments and suggestions. Please let us know if there are further concerns.

---

### Official Review · Reviewer_xkbJ · 2023-10-23

**Soundness:** 1 poor
**Presentation:** 3 good
**Contribution:** 1 poor
**Rating:** 3
**Confidence:** 5

**Summary:**

This paper considers the fairness criterion known as separation in the regression setting.  It proposes using KDE to audit/measure the violation of separation from finite samples (recall that separation involves conditioning on $Y\in\mathbb R$, whose values may be unique in a given samples).  Then, it extends the instance reweighting technique previously proposed for achieving fairness in the classification setting to regression, also via KDE (the author also consider binning and radius neighbors).

**Strengths:**

- The motivation is clear, and the criterion of separation in regression problem receives relatively less attention in the fairness literature.
- The presented material is clear and easy-to-follow.

**Weaknesses:**

1. This paper uses KDE to deal with difficulties in using finite samples to achieve fairness, but does not provide any analysis for the proposed procedures for auditing and bias mitigation.

	- What is the convergence rate of $\hat r_\mathrm{sep}$ to $r_\mathrm{sep}$ (eqs. 4 and 7)?
	- Similarly, what is the convergence rate of $\widehat W(A,Y)$ to $W(A,Y)$ (eq. 9), when estimated and inferred from finite samples using KDE/radius neighbors/binning?

2. Although an intuition is provided for the proposed FairReweighting procedure (in paragraphs surrounding eq. 8), critically, no proof or guarantees is provided for it.  Why should I trust that a regressor trained using FairReweighting would satisfy separation?

	- Under what assumptions would a regressor trained using FairReweighting satisfy separation?  Beyond toy examples?  Would it depend on the capacity/expressiveness of the regressor, or the optimization algorithm?
	- It is known that overparameterized neural networks can fit arbitrary training examples, and the effects of instance reweighting is empirically observed to be weakened [1].  Would the proposed procedure still work?
	- Is there a tradeoff between performance and fairness?  Such tradeoffs have been observed in fair classification via reweighting [2].
	- I expect some theorem upper bounding $r_\mathrm{sep}$ of regressors trained using the FairReweighting procedure, likely involving the complexity of the regressor and the number of samples.

3. Claims that the proposed FairReweighting is "free of parameters and tuned automatically" are wrong.  KDE involves specifying the variance of the Gaussian bumps, and radius neighbors needs to provide the radius and metric.

[1] Bryd and Lipton.  What is the Effect of Importance Weighting in Deep Learning?  ICML 2019.
[2] Han et al.  Balancing out Bias: Achieving Fairness Through Training Reweighting.  EMNLP 2022.

**Questions:**

See weaknesses.

---

> ### Author Response · Authors · 2023-11-22
>
> First, we would like to thank the reviewer for the detailed comments. We have carefully considered all the comments and we believe that some of the comments have driven us to improve the paper greatly.
>
> Weakness 1.
> 1) The metric $\hat{r}_{sep}$ is adopted from Steinberg et al. (2020), due to the space limitation, we will kindly refer the readers to the original paper for more details regarding this metric.
>
> 2) KDE and radius neighbors are both commonly used density estimation methods. This is why we did not analyze their convergence in this paper. Instead, we provided an empirical result in Figure 1 showing how close the density estimations are compared to the actual distributions on a synthetic dataset where the actual distributions are known.
>
> Weakness 2a. We would like to thank the reviewer for pointing out that there were not enough details on how the scheme that we introduced is related to separation. This is why we have added a new Section 3.1 to show how FairReweighing ensures separation of the learned model under a conditional independence assumption of $X\perp A | Y$. We also demonstrate the effectiveness of our approach through empirical evidence on real world datasets.
>
> Weakness 2b. In this paper, we will not be able to add results with deep neural networks due to space and time limitation. However, we do have some preliminary results for applying FairReweighing with an VGG-16 model on an image regression problem.
>
> | Pre-processing    | MAE |  R-Squared | $\hat{r}_{sep}$ |
> | -------- | ------- |  -------- | ------- |
> | None  | 0.585    | 0.622 | 1.05 |
> | FairReweighing |   0.598  | 0.603 | 1.00 |
>
>
> The above preliminary result suggests that
>  1) the effects of instance reweighting are in fact weakened by deep neural networks— without FairReweighing,  $\hat{r}_{sep}$ is already close to 1.0;
> 2) applying FairReweighing still improves separation— $\hat{r}_{sep}$ is even better after applying FairReweighing.
>
> Weakness 2c. There’s indeed a trade-off between model performance and fairness. However, as a pre-processing mechanism that is free of trade-off or “Price of Fairness” parameters (weights)--- there is no threshold or parameter to tune for the trade-off. As shown in Section 3.1, FairReweighing could achieve perfect separation under the assumption $X\perp A | Y$. Note that Reweighing is the same in terms of the trade-off.
>
> Weakness 2d. We shown Section 3.1 that FairReweighing could achieve perfect separation on the training data under the assumption $X\perp A | Y$. So, when the assumption is valid, $\hat{r}_{sep}=1.0$ on the training data.
>
> Weakness 3. The statement is indeed incorrect and we rephrased it into “free of trade-off or Price of Fairness parameters (weights)”
>
> Again, we want to thank the reviewer for the detailed comments and suggestions. Please let us know if there are further concerns.

---

### Official Review · Reviewer_7rTx · 2023-10-27

**Soundness:** 1 poor
**Presentation:** 2 fair
**Contribution:** 2 fair
**Rating:** 3
**Confidence:** 5

**Summary:**

The authors describe a technique for achieving group fairness in regression tasks. Their approach is a simple preprocessing technique that weights samples based on the observed joint distribution of protected attributes and the target variable, in order to nudge the model trained on the weighted dataset towards satisfying the standard separation fairness constraint $\hat{Y} \perp A \mid Y$, where $A$ is the protected attribute, $Y$ the target variable, and $\hat{Y}$ the model's prediction. In several simple tabular test cases, the proposed method compares favorably with other fair regression techniques.

**Strengths:**

Regression fairness has received only negligible attention compared to classification fairness, and any progress on this important topic is highly welcome. The presented approach is very straightforward - that being a good thing -, easy to implement, and very general and widely applicable to all kinds of models and training schemes. The paper is well-written and easy to read.

**Weaknesses:**

As outlined above, I consider the topic important and the solution proposed by the authors straightforward and generally promising. I do, however, sadly see quite a few significant weaknesses in the present manuscript.

Firstly, I am honestly quite confused about what it is that the authors actually implemented, and how that matches the textual description. Sections 1 and 2 suggest strongly that the authors implement a scheme to achieve separation, i.e., $\hat{Y} \perp A \mid Y$. However, the weighting scheme (section 3) seems to me to be constructed such that $p(y \mid a_1) = p(y \mid a_2)  \forall y$ - which has nothing to do with separation (note that the algorithm's predictions do not even occur) and, instead, creates a synthetic (reweighted) dataset that fulfills *demographic (or statistical) parity*, i.e., $Y \perp A$? It is completely unclear to me how this would help achieve separation for the classifier trained on such a reweighted dataset. (Note that, in any case, reweighting a training set, while possibly empirically useful, can never provide any *guarantees* about the resulting classifier having a certain property.)

Secondly, it might have been easier for me to infer what the method is actually doing if the experimental results were analyzed more comprehensively and the experiments and metrics better described. How do the trained models perform on the two protected groups separately in all of the experiments? What are the "Convex", "AOD" and "DP" metrics? (I am sure the latter is a demographic parity-related metric, but how exactly is this computed?) Why is R-Squared so abysmally low in the Law School dataset; are these all typos? What are the base rates $p(y \mid a)$ and TPR/FPR in the classification test cases? What are the actual regression models being fit; is it linear regression? Also, for the classification cases, standard equalized odds / equal opportunity classification fairness techniques as per Hardt. et al. should be included as baselines.

Thirdly, there are various statements throughout the manuscript that convey a shallow conception of "fairness" and "bias". To provide a few specific examples:
- "Separation [...] promises that a perfect predictor will always be considered the most unbiased." Separation doesn't "promise" anything. Also, "perfect" predictors can be completely biased if the target variable is noisy or biased, the dataset suffers from sampling biases, etc. Cf. e.g. Petersen et al. for a recent discussion of these issues.
- "The underlying assumption is that the unfairness we observed in machine learning models already includes discrimination rooted in training data." While data is an important source of bias, it is not the only one. Cf. e.g. Hooker and Hall et al.
- "To ensure a dataset is unbiased, the sensitive characteristics A have to be statistically independent of the ground-truth label Y. [...] Such discrepancies would lead to a biased regressor or classifier." Again, this describes a notion of statistical / demographic parity on the dataset level, not separation. Is a breast cancer dataset in which women have breast cancer more often than men "biased"? And, equivalently, is a classifier which predicts breast cancer more often for women than for men "biased"?

Fourthly, given that this is an immediate application of widely used standard techniques, the discussion of and references to prior relevant work are quite sparse.
- Weighting and over/undersampling techniques are widely used throughout the algorithmic fairness literature, see e.g. Caton and Haas.
- This is essentially an application of covariate shift adaptation techniques as per Sugiyama et al., with the "target population" being the one where a desired notion of group fairness is satisfied.
- While prior work on fair regression is discussed quite well, there are a few further methods that might be relevant to mention, such as Komiyama et al. and Calders et al. In particular Calders et al. use a propensity score-based approach, which is very closely related to what the authors are describing here; the differences should be spelled out clearly.

**References**
- Calders et al. (2013), Controlling Attribute Effect in Linear Regression, https://ieeexplore.ieee.org/document/6729491
- Caton and Haas (2023), Fairness in Machine Learning: A Survey, https://doi.org/10.1145/3616865
- Hall et al. (2022), A Systematic Study of Bias Amplification, https://arxiv.org/abs/2201.11706
- Hooker (2021), Moving beyond “algorithmic bias is a data problem”, https://doi.org/10.1016/j.patter.2021.100241
- Komiyama et al. (2018), Nonconvex Optimization for Regression with Fairness Constraints, http://proceedings.mlr.press/v80/komiyama18a.html
- Petersen et al. (2023), The path toward equal performance in medical machine learning, https://www.cell.com/patterns/fulltext/S2666-3899(23)00145-9

**Questions:**

- The authors correctly point out that direct density estimation is known to be more efficient than estimating the two likelihoods separately (a reference on this statement would also be appropriate) - and then they proceed to estimate the two likelihoods separately and do *not* do direct density estimation?
- I believe Eq. (4) is missing an expectation? (Currently, it does not evaluate to a number.)
- I believe in Eq. (7), the ratio is inverted compared to Eq. (4)?
- Which of the two discussed density approximation methods is used for the results in Fig. 1, and for the further results? Also, "this suggests that our methods can correctly estimate the density of any given variable" is a very strong claim given that the method was evaluated on exactly one very simple test case.
- What is "Models = Ground Truth" in Table 1?
- I assume the tau in all tables should be a tauhat?
- Where can I find the results for the described experiments with continuous protected attributes?
- The authors write that "FairReweighting" is "essentially equivalent" to the previously proposed Reweighting method - isn't it precisely, mathematically, identical?

---

> ### Author Response · Authors · 2023-11-22
>
> First, we would like to thank the reviewer for the detailed comments. We have carefully considered all the comments and we believe that some of the comments have driven us to improve the paper greatly.
>
> Weakness 1. We would like to thank the reviewer for pointing out that there were not enough details on how the scheme that we introduced is related to separation. This is why we have added a new Section 3.1 to show how FairReweighing ensures separation of the learned model under the condition that $X\perp A | Y$. We also demonstrate the effectiveness of our approach through empirical evidence on real world datasets.
>
> Weakness 2. We apologize for the confusions raised in the previous experiment descriptions. We have substantially revised the experiment section:
>
> 1) We have removed fairness metrics other than $\hat{r}_{sep}$ from Table 1 and 2 to avoid confusions.
> 2) We explained that the base models for FairReweighing are linear regression for regression and logistic regression for classification in the first paragraph of Section 4.
> 3) We also explained the baselines compared in Table 2 in the first paragraph of Section 4 and in Section 4.1.3.
> 4) We clarified in Section 4.2.2 that the experiment on the classification data is just to show that FairReweighing is the same as Reweighing in classification. And that $\hat{r}_{sep}$ can evaluate separation as good as AOD— average odds difference (a metric used to evaluate equalized odds) in classification. This is why we did not analyze too much for the classification results— the Reweighing paper already did that. The conclusion we draw from this experiment is that FairReweighing can be applied to classification and when it is applied to classification, it is the same as Reweighing. This validates our claim that FairReweighing is a generalized form of Reweighing on regression problems.
>
>
> Weakness 3. Thanks for pointing these out! We have rephrased multiple statements throughout the manuscript so it would be more accurate and correctly reflect the conception of "fairness" and "bias".
>
> Weakness 4. We greatly appreciate the literature that the reviewer recommended. After careful selection, we include those that we believe could complement our discussion and comparison with prior relevant works.
>
> Question 1. We have clarified in Section 3.2 that the “direct estimation” only applies to classification problems. In regression problems, we have to use the nonparametric probability density estimation techniques for regression problems.
>
> Question 2, 3 & 6. Fixed
>
> Question 4. We rephrased the claim into an observation of the results.
>
> Question 5. “Ground Truth” refers to implementing FairReweighing with the actual ground truth distribution (we know this because it is a synthetic dataset) instead of density estimation approximation. It was confusing and we added an explanation in the result analysis in Section 4.1.1.
>
> Question 7. Currently we are only evaluating our approach’s effectiveness on binary protected attributes upon which separation is defined. We have specified this in the problem statement of Section 3. It’s definitely one of the major directions of our future work to generalize it to continuous sensitive attributes.
>
> Question 8. The major contribution of this work is a generalization of Reweighing (Kamiran & Calders) onto regression problems with k-nearest neighbors or kernel density estimation. The math is indeed identical with the previously proposed Reweighting but not how we calculate the weight. We also demonstrated in RQ2 that when applied to classification problems, FairReweighing is identical to Reweighing.
>
> Again, we want to thank the reviewer for the detailed comments and suggestions. Please let us know if there are further concerns.

---

### Official Review · Reviewer_NWYi · 2023-10-30

**Soundness:** 3 good
**Presentation:** 4 excellent
**Contribution:** 3 good
**Rating:** 6
**Confidence:** 4

**Summary:**

The author proposes a density estimation-based preprocessing algorithm to train regression models, with the goal of reducing the bias of the original data before the entire training process. This method is more efficient and has lower overhead. This method has no parameters and can be automatically tuned to achieve a balance between all specified protected attributes. The experimental results also indicate that the algorithm proposed by the author improves separation in fair regression while maintaining high prediction accuracy, and its performance is superior to the most advanced existing schemes.

**Strengths:**

The paper includes a comprehensive summary of related work. The ideas of the paper are clearly presented. The author proposes a universal preprocessing framework, which is a training framework based on density estimation. By adjusting the impact of each data item through weight allocation to achieve fairness, classification and regression models can be effectively trained. This article extends the fairness issue in classification problems to regression problems based on previous studies.

**Weaknesses:**

Lack of analysis on the robustness and stability of the algorithm: The paper did not analyze the robustness and stability of the algorithm. In practical applications, algorithms need to be able to handle various uncertainties and noise while maintaining stable performance. The analysis of the robustness and stability of algorithms can provide a more comprehensive evaluation.

**Questions:**

1. The paper did not provide specific details and implementation methods for the kernel density estimation algorithm and lacked transparency in the algorithm to verify its effectiveness.
2. Lack of comparison with other methods: There are relatively few existing classification fairness methods in the paper, making it difficult to determine the advantages and disadvantages of this method in classification tasks.
3. The paper mentioned some fairness measures but did not provide a detailed discussion on the selection and applicability of these measures. The selection of fairness measures is crucial for evaluating the fairness of algorithms and requires more in-depth discussion and explanation.
4. The paper provides evaluation results for both synthetic and real-world data but does not provide validation for practical application scenarios. The data in practical application scenarios may be more complex and diverse and is the weight allocation method proposed in the paper still effective in improving fairness.

---

> ### Author Response · Authors · 2023-11-22
>
> First, we would like to thank the reviewer for the detailed comments. We have carefully considered all the comments and we believe that some of the comments have driven us to improve the paper greatly.
>
> Question 1. We acknowledge the lack of details on how kernel density estimation is implemented in our experiment and hence added more information at the start of Section 4. Figure 1 also shows the estimation accuracy of kernel density estimation on synthetic data where we know the ground truth probabilities.
>
> Question 2. The contribution of this paper focuses on the extension of both fairness metrics and bias mitigation techniques from binary classification tasks to regression problems. Therefore we have focused on the experiments on regression tasks and the comparisons against other regression-friendly algorithms. The experiments and comparison on classification is merely a demonstration of the generalizability of our method in both situations. Also note that on the classification data, FairReweighing is the same as Reweighing and Separation can be evaluated by equalized odds.
>
> Question 3. We have provided a more thorough discussion on the selection and applicability of all the fairness measures we are comparing at the end of Section 2.2.1. In short, we focus on the Separation criterion only in this paper since it always allows the perfect predictor to be evaluated as fair while individual fairness requires similar individuals (from different sensitive groups) to be treated similarly, demographic parity requires the acceptance rates to be the same across different sensitive groups.
>
> Question 4. Following the suggestion from the reviewer, we have added descriptions (highlighted in Section 4.1.2) for the potential practical application scenarios for our two real world data.
>
> Again, we want to thank the reviewer for the detailed comments and suggestions. Please let us know if there are further concerns.

---

### Author Response · Authors · 2023-11-22
**Global Author Rebuttal**

We would like to thank the reviewer for the detailed comments and suggestions. We have revised the paper accordingly and the most important change is the newly added Section 3.1 explaining why FairReweighing as a pre-processing step would help satisfying the separation criterion in regression problems. Questions about this have been raised by almost every reviewer and that is why we carefully explained that in Section 3.1. There are also other clarifications being made all over the revised paper and all the changes are highlighted in blue. We hope that the revised version has addressed all of the previous concerns, and we will also greatly appreciate the reviewers to continue to provide feedback on the revised version.

Thank you!

---

### Meta-Review · Area_Chair_SbQB · 2023-12-09

**Metareview:**

The paper introduces a pre-processing algorithm based on density estimation for training "fair" regression models. The algorithm aims to ensure a "separation" condition where, conditioned on the true outcome $Y$, the predicted outcome  $\hat Y $ is independent of a group attribute $A$. The reviewers appreciated the summary of related work and noted that the proposed approach performs well in experiments. However, several reviewers raised concerns regarding the theoretical analysis of the method, noting that further theoretical guarantees on FairReweighing could be given.

**Justification For Why Not Higher Score:**

Though promising, the reviewers found that the limitations of the paper outweigh its merits. Further revision is required prior to acceptance.

**Justification For Why Not Lower Score:**

N/A

---

### Decision · Program_Chairs · 2024-01-16

Reject